# Standardization of Cell Culture Conditions and Routine Genomic Screening under a Quality Management System Leads to Reduced Genomic Instability in hPSCs

**DOI:** 10.3390/cells11131984

**Published:** 2022-06-21

**Authors:** Francisco J. Molina-Ruiz, Clelia Introna, Georgina Bombau, Mireia Galofre, Josep M. Canals

**Affiliations:** 1Laboratory of Stem Cells and Regenerative Medicine, Department of Biomedical Sciences, Faculty of Medicine and Health Sciences, University of Barcelona, 08036 Barcelona, Spain; fjmolina@ub.edu (F.J.M.-R.); c.introna@ub.edu (C.I.); gbombau@ub.edu (G.B.); mireiagalofre@ub.edu (M.G.); 2Creatio, Production and Validation Center of Advanced Therapies, Faculty of Medicine and Health Science, University of Barcelona, 08036 Barcelona, Spain; 3Institute of Neurosciences, University of Barcelona, 08036 Barcelona, Spain; 4Networked Biomedical Research Centre for Neurodegenerative Disorders (CIBERNED), 08036 Barcelona, Spain; 5August Pi i Sunyer Biomedical Research Institute (IDIBAPS), 08036 Barcelona, Spain

**Keywords:** human pluripotent stem cells, genomic instability, karyotype, cell therapy, ISO9001, GIVIMP

## Abstract

Human pluripotent stem cells (hPSCs) have generated unprecedented interest in the scientific community, given their potential applications in regenerative medicine, disease modeling, toxicology and drug screening. However, hPSCs are prone to acquire genomic alterations in vitro, mainly due to suboptimal culture conditions and inappropriate routines to monitor genome integrity. This poses a challenge to both the safety of clinical applications and the reliability of basic and translational hPSC research. In this study, we aim to investigate if the implementation of a Quality Management System (QMS) such as ISO9001:2015 to ensure reproducible and standardized cell culture conditions and genomic screening strategies can decrease the prevalence of genomic alterations affecting hPSCs used for research applications. To this aim, we performed a retrospective analysis of G-banding karyotype and Comparative Genomic Hybridization array (aCGH) data generated by our group over a 5-year span of different hESC and hiPSC cultures. This work demonstrates that application of a QMS to standardize cell culture conditions and genomic monitoring routines leads to a striking improvement of genomic stability in hPSCs cultured in vitro, as evidenced by a reduced probability of potentially pathogenic chromosomal aberrations and subchromosomal genomic alterations. These results support the need to implement QMS in academic laboratories performing hPSC research.

## 1. Introduction

Pluripotent stem cells (PSCs) are a unique class of undifferentiated cells characterized by an unlimited self-renewal capacity and their ability to differentiate into all specialized cells of the adult body by developing into cell types that represent the three embryonic germ layers—mesoderm, ectoderm and endoderm. PSCs can be classified into two main types with different origins: embryonic stem cells (ESCs) and induced pluripotent stem cells (iPSCs). On one hand, ESCs are derived from the inner cell mass (ICM) of preimplantation embryos [1,2], whereas the iPSCs are produced by inducing dedifferentiation of adult somatic cells via cell reprogramming [3,4]. These two cell types, which share the two aforementioned hallmark properties of self-renewal and pluripotency, hold unprecedented promise for a wide range of uses, including mechanisms of embryonic development [5], disease modelling, drug discovery [6], and regenerative medicine [7]. For instance, although it has only been 24 years since human ESCs (hESCs) were first derived [2], and around 15 years since human iPSCs (hiPSCs) were originally described [4], the use of human PSCs (hPSCs) has gained momentum and become increasingly widespread. At a research level, these cells can be a valuable tool to gain insight into the mechanisms underlying human organogenesis and enhance understanding of disease mechanisms. Moreover, they can serve as an alternative in vitro platform for drug and toxicology testing to bypass the need for animal models and thus avoiding issues related to failed translation to humans as well as the ethical concerns surrounding animal research [8], hPSC-based therapies for the treatment of degenerative diseases including heart disease, diabetes or neurodegenerative disease have been developed by the differentiation of hPSCs to the affected cell type for subsequent transplantation into patients. In fact, clinical trials for regenerative medicine using derivatives of these cells are already underway or are planned and have, up to now, shown the cells to be generally safe [9,10,11,12,13,14,15].

Despite the progress and the investment made towards research and development aimed at harnessing the curative potential of hPSCs, their clinical translation has been hampered by the presence of mutations in these cells. Although hPSCs are mostly diploid when first derived, it is now recognized that they can acquire different types of genetic alterations on prolonged cell culture passages [16]. Indeed, one potential clinical trial of retinal pigment cells from an autologous iPSC line for age-related macular degeneration was halted once it was discovered that the cells carried a point mutation even though the significance of this particular mutation was unknown [9,17]. This case highlighted the fact that genetic changes in hPSCs could jeopardize not only the safety of hPSC-based cell products for regenerative medicine [18,19], but also lead to heterogeneous differentiation propensity of the starting material, altered gene expression profiles and inefficiency of the final cell product which would also affect other applications, such as disease modelling and drug discovery [16].

Around twenty years ago, genomic instability in hPSCs was first recognized when karyotypic abnormalities in hESCs were detected by G-banding karyotypes as non-random gains of particular chromosomes or fragments of chromosomes, notably chromosomes 12 and 17 [20,21]. Later on, a large collaborative study published in 2011 by the International Stem Cell Initiative (ISCI) showed that these aberrations were present across the entire genome in cell lines analyzed worldwide [22]. As of today, many different genomic abnormalities have been discovered in both hESC and hiPSC lines [23,24,25], ranging from whole chromosome aneuploidies to subchromosomal aberrations (such as gene duplications and deletions and single base pair changes) [22,26,27,28]. Interestingly, many of these genetic abnormalities are often recurrent, and while aberrations are found across the entire genome, there are some chromosome regions more prone to gains or losses. For instance, the most common genomic abnormalities present in hPSCs include chromosome 1 or 1q amplification, trisomy 12 or 12p amplification, trisomy X (female cell lines), trisomy 17 or 17q amplification and 20q11.21 amplification [22,24,29,30,31,32,33]. They represent the majority of the karyotypic abnormalities found in hPSCs and could thus be defined as recurrent, followed in terms of frequency by losses of chromosomes 18q and 10p which occur to a lesser extent [22]. Interestingly, some of these recurrent abnormalities are particularly subtle and difficult to detect by most widely used standard cytogenetic procedures (G-banding), and may therefore go unnoticed even when present in a substantial proportion of the cells [16].

The repetitive, non-random nature of many of these recurrent mutations observed in hPSCs, as well as their high prevalence and its propensity to rapidly out-compete normal cells in culture within few passages, has been directly linked to the selective growth advantage that these aberrations confer to the cells in culture [34]. Furthermore, genetic instability may not only provide a selective advantage in the form of a faster cell cycle time, an altered pattern of differentiation, or an increased rate of survival, but also change hPSC functional characteristics, including decreased differentiation potential, increased capacity to self-renew and a shift towards malignant properties. In fact, many of the recurrent chromosomal aberrations seen in hPSC are also associated with known human cancers, which raises concerns over using these cells for therapeutic applications. Overall, due to their frequency and functional consequences which can affect the outcome of both research, and clinical applications recurrent genetic alterations found in cultured hPSCs are not acceptable. Even if the underlying mechanisms that cause the observed genetic variations are unclear, it has become evident that suboptimal in vitro culture condition [35] and the high levels of DNA damage compared to somatic cells are the two main drivers of genomic alterations which compromise the maintenance of genetically normal hPSCs. Consequently, and even if some of the abnormalities detected in hPSCs are likely to have been already present in the embryos or somatic cells from which the cell line was derived or have been induced during such derivation [22,36], an increasing number of reports link the occurrence of acquired genomic abnormalities to suboptimal cell culture techniques, prolonged passage or changing the culture conditions [20,37,38,39].

In order to ensure the genomic integrity of hPSCs, it is of foremost importance to develop culture conditions that minimize change in hPSC cultures. Implementing appropriate screening methods which can reliably detect not only chromosomal aneuploidies, but also more insidious subchromosomal aberrations is as important as minimizing genomic variability in hPSC cultures. However, and mainly due to the cost and labor involved, current screening practices are infrequent and unsatisfactory, which could lead to the use of hPSCs harboring undetected mutations which could affect their research and clinical applications. Besides wasted time and resources, the inability to detect genomic alterations in hPSCs can result in the generation of erroneous data in the literature which could both confuse and delay scientific progress in this area. Along these lines, the genetic instability showed by hPSCs in culture makes the frequent reassessment of genomic integrity an essential requirement when planning to use them for experiments, as an increasing number of reports are recommending [7,15].

As clinical trials using hPSC-derived cells are underway, it is crucial that we understand the implications of genomic alterations in hPSCs, and how to detect and minimize their occurrence in hPSC cultures. To accomplish this goal, we need to ensure reproducible conditions of maintenance and genomic screening. Thus, the implementation of a quality management system (QMS) which defines standards for hPSC culture, controls, detects, and tracks errors affecting the quality of the cultured cells, and ensures the accuracy, reliability, and traceability of results is indispensable not only to minimize genomic variability in hPSC in vitro, but also to avoid misdetection of genomic alterations of the cell lines. Under a QMS, changes in cell culture conditions are minimized, cell culture practices are optimized, and the quality of the used reagents is ensured, which provides a fertile ground to explore its effects on the genetic stability of hPSCs. In fact, the Organization for Economic Co-operation and Development (OECD) has recently published the Guidance Document on Good In Vitro Method Practices (GIVIMP) [40] to foster confidence in in vitro alternatives to animal testing. The development and use of standard operating procedures (SOPs), which are a set of written instructions that document a routine or repetitive activity (i.e., cell culture protocols or procedures for genome integrity assessment) followed by an organization, are an integral part of a successful QMS such as ISO9001:2015. Any given SOP provides individuals with the information to perform a job properly in the most detailed possible fashion, thus minimizing variation and promoting quality through consistent implementation of all processes or procedures within the organization. In the present study, we investigate if the introduction of SOPs under ISO9001:2015 for standardized cell culture conditions and frequent assessment of genomic stability on hPSC lines can decrease the prevalence and types of genetic changes affecting hPSCs used for research applications. To this aim, we carried out a retrospective analysis of data generated from conventional G-banding karyotype and array-based Comparative Genomic Hybridization (aCGH) analysis over a 5-year span (Jan 2017–Jan 2022) on hESC and hiPSC cultures maintained by our research group, before and after adaptation to standard conditions defined by appropriate SOPs under a QMS. This work shows that adaptation of hPSC lines to standardized cell culture conditions under a QMS leads to reduced genomic instability of hPSCs in vitro, as evidenced by a reduced probability of chromosomal aberrations and subchromosomal genomic alterations in hPSC lines.

## 2. Materials and Methods

### 2.1. Literature Search

We used NIH’s PubMed.gov (https://pubmed.ncbi.nlm.nih.gov/ (accessed on 24 February 2022) to perform several literature searches of published studies of interest. For all of them, we excluded from the search all studies which were published before 1990 or after 2021. We filtered publications of interest using the following search queries: (i) hPSCs, hPSC, human pluripotent stem cell or human pluripotent stem cells; (ii) hESCs, hESC, human embryonic stem cell or human embryonic stem cells; (iii) hiPSCs, hiPSC, human induced pluripotent stem cell or human induced pluripotent stem cells; (iv) (hPSCs, hPSC, human pluripotent stem cell or human pluripotent stem cells) and (karyotype, G-band or aCGH); (v) (hESCs, hESC, human embryonic stem cell or human embryonic stem cells) and (karyotype, G-band or aCGH); and (vi) (hiPSCs, hiPSC, human induced pluripotent stem cell or human induced pluripotent stem cells) and (karyotype, G-band or aCGH).

### 2.2. Cell Lines

We analyzed a total of 11 control and 8 disease-affected female hPSC lines, which are listed in Table 1. All the disease-affected hPSC lines carry an Htt gene CAG expansion of different length, but were indicative of Huntington’s disease in all cases. The use of hPSC lines of the present study is under the ethical permission 0336/2939/2019 (Generalitat de Catalunya, Barcelona, Spain).

### 2.3. Initial Expansion of hPSC Lines

We received the hPSC lines from several centers and the cells were initially expanded using the original protocols [Pre-adaptation (PrA) condition]. We tested up to 5 different protocols which have been previously described in the literature [41,42,43,44,45].

### 2.4. Human PSC Maintenance in mTeSR1 Media with ROCK Inhibitor

A vial of the initial PrA cell bank was frozen and expanded using hPSC lines to the standard cell culture conditions used in our laboratory [Post-adaptation (PoA) condition] refer to hPSCs analyzed before and after completing adaptation to, respectively. All hPSCs were cultured on Matrigel (354230; Corning Inc., New York, NY, USA) feeder-free coated 6-well plates in 2 mL of serum-free defined mTeSR1 medium which contains recombinant human bFGF and recombinant human TGFβ (complete mTeSR1: mTeSR1 basal medium (85850) and mTeSR1 supplemented 5 × 8 (85852); StemCell Technologies Inc., Cambridge, MA, USA). The medium was changed every day (every 24 h ± 2).

The hPSC cultures were passaged when cell confluence reached 70–80% (twice per week). Cells were gently washed twice with 2 mL of Dulbecco’s (D)-phosphate-buffered saline (PBS) without calcium and magnesium (D-PBS−/−, 14190–250; Thermo Fisher Scientific Inc., Waltham, MA, USA) and incubated for 20–25 min with dispase in DMEM/F12 (07923; StemCell Technologies Inc., Cambridge, MA, USA) with 10 μM of ROCK inhibitor (Y-27632; EVOTEC Ltd., Milton, UK) at 37 °C. Then, dispase was removed, and cells were harvested in DMEM/F12 (36254; StemCell Technologies Inc., Cambridge, MA, USA) media by pipetting up and down using a 5-mL pipette to avoid single-cell desegregation. Cells were centrifuged at 1250 rcf for 3 min and resuspended with warm and fresh Complete-mTeSR1 medium with 10 μM of ROCK inhibitor at the desired dilution. Then, cell suspension was seeded on tempered Matrigel coated 6-well plates. Differentiated colonies were removed manually using a stereotypic microscope and a 10-mL sterile tip prior passaging.

### 2.5. Sample Fixation for Karyotyping (G-Banding)

The hPSCs were karyotyped regularly upon thawing and after every 10 passages, as well as before and after experimental use of hPSCs (as to ensure genomic integrity of the starting material) at Ambar Lab (L’Hospitalet de Llobregat, Barcelona, Spain) by G-banding test. Nevertheless, samples were prefixed to preserver the samples until their analysis. Three wells of six-well plates at 60% confluent were raised as follows. Cultures were incubated for 1 h with Complete mTeSR1 with 0.2 µg/mL of Karyomax Colcemid (15212-012; Life Technologies, Vancouver, Canada) at 37 °C. Then, cultures were washed thrice with 2 mL of D-PBS−/− and incubated for three minutes with StemPro Accutase (A1110501; Thermo Fisher Scientific Inc., Watham, MA, USA) at 37 °C. Then, cells were desegregated as a single-cell, and 4 mL of complete mTeSR1 were added per well to dilute and inactivate the enzyme. Cell suspension was transferred to a 15 mL conical tube and centrifuged for 3 min at 1250 rcf. Then, supernatant was aspirated carefully to not aspirate the pellet of cells which were washed with 5 mL of D-PBS−/− and centrifuged again for 3 min at 1250 rcf. The pellet was gently resuspended in 10 mL of hypotonic solution composed of 0.075 M of KCl (10575-090; VWR International Eurolab, Barcelona, Spain). The hypotonic solution was added gently drop-to-drop while the tube was agitated. Cell suspension was kept for 10 min at 37 °C water bath to rest and then pre-fixed with 1 mL of cold (−20 °C) Carnoy fixative solution composed of (1:3) Methanol (20847; VWR International Eurolab, Barcelona, Spain) and Acetic Acid (1310081612; PANREAC Quimica SLU, Barcelona, Spain). Carnoy was gently dropped while vortexing the tube. After 10 min of centrifugation at RT and 2220 rcf, pellet was resuspended in 10 mL of cold (−20 °C) Carnoy fixative solution while vortexing. The tube was sealed and stored at −20 °C until picked up for its analysis.

### 2.6. Sample Preparation for Microarray-Based Comparative Genome Hybridization (aCGH)

The aCGH assay was performed regularly before and after adaptation to standardized cell culture conditions detailed in appropriate SOPs, by Ambar Lab. A cell suspension is harvested in the laboratory before sending the sample to the company. The hPSC cultures were harvested when cell confluence reached 80%. Cells were gently washed twice with 2 mL of D-PBS−/− and incubated for 5 min with StemPro Accutase at 37 °C. Then, single-cell suspension was harvested in 2 mL of complete mTeSR1. Cells were centrifuged at 1250 rcf for 3 min and resuspended with 2 mL of D-PBS−/− and set at 4 °C before the immediate shipping to Ambar Lab.

### 2.7. Alteration Probability Analysis

The alteration probability analysis was performed using a Kaplan–Meier survival analysis. The hPSCs which did not have an event occurrence (detection of the studied genomic alteration subtype) were censored. The hPSCs were divided into two different groups according to whether they had been adapted to standardized in vitro cell culture conditions under a QMS. Adaptation was considered complete when at least 4 passages had been performed under these conditions. Passage number for both groups was normalized before analysis.

### 2.8. Statistical Analysis

Statistical analysis was carried out using GraphPad Prism version 8.0.1 (GraphPadSoftware Inc., San Diego, CA, USA). Data are presented as mean ± the standard error of the mean (SEM). Statistical analysis of continuous numerical data was performed using a two-tailed Student’s t-test. Survival analysis was performed using a Kaplan–Meier survival analysis and statistical significance was determined by a log-rank (Mantel–Cox) test. For all statistical analyses, *p* values were computed using the NEJM (New England Journal of Medicine) style. Differences were statistically significant at *p* < 0.05. One asterisk (*) summarizes *p* values less than 0.05 (significant), two asterisks (**) summarize *p* values less than 0.01 (very significant) and three asterisks (***) summarize *p* values less than 0.001 (extremely significant). *p* values above 0.05 (non-statistically significant) are flagged with “ns”.

### 2.9. Quality Management System (QMS)

All procedures were conducted under UNE-EN-ISO9001:2015 and comply with the Guidance Document on Good In Vitro Method Practices (GIVIMP; OECD) recommendations [40].

## 3. Results

### 3.1. Analysis of the Evolution in Number of hPSC-Related Publications and Level of Reporting of Genomic Integrity Monitoring

#### 3.1.1. The Number of Publications on hPSCs per Year Has Experienced a Large Increase during the Last 20 Years

We first investigated the evolution of the scientific interest on hPSC over time, using the number of scientific publications on hPSCs per year as an indicator. To that aim, we performed a literature search worldwide to quantify the number of scientific publications on hPSCs during the last 32 years (January 1990–December 2021). Our sampling procedure yielded 44464 records of scientific publications, whose distribution over the studied period was heterogeneous (Figure 1A). Briefly, we observed a 56-fold increase in the number of hPSC publications during the studied period. We found that while the number of papers published per year on hPSCs initially remained stable, it started to increase regularly around 1998, when 134 papers were published. In fact, by 2007, published research works on hPSCs had increased more than 5-fold since 1998 (1187 publications). From then, the increase becomes steeper, and we observe an exponential rise in scientific works published per year on hPSCs, which has been maintained until the 4130 publications on hPSCs during 2021 (more than a 30-fold difference compared to 1998 and a 3.5-fold difference compared to 2007). We hypothesized that the inflection points observed in 1998 and 2007 could be explained by the fact that two of the most important breakthroughs in the hPSC field were reported in those years, namely the first derivation of an hESC line by Thomson et al. [2] and the first generation of an hiPSC line by Yamanaka et al. [4], respectively.

To assess if differences existed among these hPSC types regarding the scientific interest that they have raised over time, we conducted two additional literature searches on the number of papers published per year during the same period on hESCs and hiPSCs separately. Our sampling procedure yielded 27325 and 26991 records of publications on hESCs and hiPSCs, respectively. We studied the distribution of these publications over time and discovered different trends depending on the hPSC type (Figure 1B). For instance, the number of papers published per year on hESCs rises from 1998 until reaching a maximum of 1822 around 2014, where interest in hESC research seems to decline; in turn, the rate of publication on hiPSC begins to rise in 2007, sometime after that of hESCs, but this increase is faster and uninterrupted until present. For instance, as of 2021, the number of publications on hiPSCs (3436) more than doubled the number of papers on hESCs (1148). Furthermore, from these results we identified 2013 as a third important inflection point regarding scientific interest on hPSC research. However, this turning point led to different trends depending on the analyzed hPSC type; while a decrease on published hESC research was observed, the number of publications per year on hiPSCs clearly increased. We hypothesized that this effect could be explained by the onset of the first clinical trial using autologous iPSC-derived cells in 2013 [46].

To further confirm the significance of these results, we used the three milestones that we previously identified as inflection points (1998, 2007 and 2014) to define four different time segments (given the influence that they have on the trend of hPSC, hESC and hiPSC-related publications per year): (i) 1990–1997 (previous to the first hPSC derivation); (ii) 1998–2005 (between the first hESC derivation and the first hiPSC generation); (iii) 2006–2013 (between the first generation of a hiPSC line and the first hiPSC-based clinical trial); (iv) 2014–2021 (after the first application of hiPSC-derived cell products in humans. We compared the average number of publications per year for each of those segments (Figure 1C). As we expected, the average number of publications per time segment increased over time. Furthermore, the differences we found between adjacent time segments were heterogeneous, which indicates that each of the events leading to an inflection point had a variable effect on the analyzed parameter. For instance, the greatest increase that we observed was an 8-fold change from 1998–2005 to 2006–2013. Hence, we identified the creation of the first hiPSC lines as the most influential event in raising interest on hPSC research during the studied period. On the other hand, differences between the other time segments remained constant at approximately 2 to 3-fold difference.

#### 3.1.2. Data on hPSC Genomic Integrity Remains Underreported despite the Fast Growth of hPSC Research

Once we confirmed the fast-growing increase in interest of hPSC research since the first hPSC derivation in 1998, we investigated whether this increase in hPSC research had been accompanied by an appropriate reporting of data on genome integrity of the hPSCs being experimentally used. To that aim, we searched the number of publications on hPSC research which contained information on genomic integrity monitoring of hPSCs during the same period (January 1990–December 2021). This time, our sampling procedure yielded only 2340 records, which represents only 5.26% of the total number of publications during that time (Figure 1D). We interpret this result as an indicator of a lack of appropriate reporting of genomic integrity of hPSC lines being used in research, despite the fast growth that the field is experiencing. Furthermore, we repeated this search again to look at the evolution of this parameter for hESCs and hiPSCs individually (Figure 1E). For both groups, we observed an initial positive evolution starting when the first hESC or hiPSC line was derived. The number of papers including genomic integrity data of hESCs kept increasing until 2010, before experiencing a decline which would result in having a similar number of publications containing genomic integrity information of both groups by 2014. From 2015 to 2016, we observed a steep 2-fold increase for both groups which matches the time period by when the RIKEN hiPSC clinical trial was halted [17]. From that point, and following the previously reported trends in hESC and hiPSC-related scientific publications during those 30 years, we observed a sudden decrease in hESC-related publications, whereas the number of hiPSC-related papers containing kept increasing steadily after a small decline.

### 3.2. Analysis of the Types and Prevalence of Genomic Alterations Acquired by hPSCs In Vitro

#### 3.2.1. Descriptive Statistics of the Analyzed hPSC Dataset

We have performed routine hPSC in vitro maintenance and expansion for over 10 years in our laboratory, despite data on genomic stability of our hPSC lines being only available from 2017. This was achieved by adapting all our activities and processes to be performed according to high quality standards under compliance of ISO 9001:2015 rules and the Guidance Document on Good In Vitro Method Practices (GIVIMP; OECD) recommendations. In order to investigate whether the implementation of these standards and consequent development and use of SOPs could lead to improved cell culture conditions and genomic screening routines and a subsequent decrease in the number of recurrent genomic alterations of hPSCs, we performed a retrospective post-hoc analysis of data generated from routine genome integrity screening hESC and hiPSC cultures maintained by our research group over the 5-year span for which data were available (January 2017–January 2022). The composition and descriptive statistics of the analyzed dataset can be found in Figure 2. Briefly, we included a total of 19 different hPSC lines, from which 11 (57.89%) are hiPSCs whereas the remaining 8 (42.11%) are hESCs (Figure 2A). These cell lines were routinely cultured, and genome integrity data were obtained using two different methods which differ in resolution: karyotypic G-banding and aCGH. In total, data from more than 120 tests performed in hESC and hiPSC lines at different passage numbers was analyzed in this study (Figure 2B,C).

#### 3.2.2. Recurrent Karyotypic Aberrations Are Detected by G-Banding with a Low Frequency

We started by investigating the types and prevalence of chromosomal abnormalities detected in our hPSCs by the classical karyotypic technique G-banding. After gathering data from 108 G-banding tests, we found that 24.1% of the hPSCs being analyzed showed genomic alteration which could be detected with the resolution of the used cytogenetic technique (that is, abnormalities above 5–10 Mb), whereas 75.9% of the hPSCs seemed to be unaltered. Furthermore, by distinguishing between pathogenic aberrations and frequent polymorphic variants, we noticed that 69.2% of the variants corresponded to the former; consequently, the pathogenic variants represented only 16.67% of all the analyzed hPSCs whereas frequent polymorphic alterations (FPAs) were discovered in 7.41% (Figure 3A).

Then, we investigated the types of chromosomal abnormalities that were detected in hPSCs by using this approach (Figure 3B and Table 2). The most abundant alteration that we found was a recurrent gain in the long arm of chromosome 20, in the form of an isochromosome 20q (more specifically i(20) (q10)). This alteration was present in 11.11% of the analyzed hPSCs. Interestingly, this was not the only gain involving chromosome 20, since we also observed trisomy of this chromosome in 0.92% of hPSCs, which means that chromosome 20 gains accounted for 50% of the total chromosomic alterations that we found (and 72.22% of all pathogenic alterations). We also detected other recurrent chromosomal gains in hPSCs, such as trisomies of chromosomes 8 (47, XX + 8) in 2.78% of hPSCs and 12 (47, XX + 12) in 1.85% of hPSCs. Interestingly, all detected gains were present in heterogeneous mosaic cultures in which the alteration was present in variable relative amounts, ranging from 10–20% for 47, XX,+8, 20% for i(20) (q10) or 25% for 47, XX, +20, to 40–50% of the cells for 47, XX,+12. Notably, all chromosomal aberrations were gains of whole chromosomes or chromosomal regions, which contrasts with the fact that chromosomal losses were not detected in any of the analyzed samples. Focusing on the non-pathogenic polymorphic variants, we found a pericentric inversion in one or both chromosomes 9 (46,XX,inv(9) (p11q13)) in 3.7% of hPSCs and the satellites in the distal region of the long arm of one of the 21 chromosomes (21qs+) in another 3.7% of hPSCs; both correspond, as reported above, with frequent polymorphic variants found in hPSCs in vitro.

#### 3.2.3. The hiPSCs Show an Increased Propensity to Acquire Chromosomic Aberrations

Next, we assessed differences in the propensity to acquire these chromosomal aberrations among different hPSC types. To that aim, we studied whether the probability of hESCs to acquire chromosomal alterations over time was significantly different than that of hiPSCs. We first normalized the passage number for both hPSC groups. We then used Kaplan–Meier curves to show the probability of hESCs and hiPSCs to acquire genomic alterations detectable by G-banding at a certain passage number (Figure 4A). The hESCs and hiPSCs were analyzed over a period of 42 passages in vitro. We found that chromosomal alterations in hESCs were significantly lower than in hiPSCs over the time (*p* = 0.03). In fact, this probability was around 2 times higher (~33% vs. ~15%) for hiPSCs by passage 42 compared to hESCs. 

Furthermore, we also assessed the absolute frequency at which chromosomal alterations were observed in hESCs and hiPSCs using G-banding (Figure 4B). Once again, we observed an increased propensity of hiPSCs to acquire these mutations, being 2.7 times more likely than hESCs to acquire chromosomal alterations detected by G-banding (the observed frequency of altered karyotypes was 36.73% and 13.56% for hiPSC and hESC lines, respectively). By limiting the analysis to prevalence of pathogenic variants, we noticed a 1.9-fold increase in hiPSC lines (22.45%) compared to hESC lines (11.86%). Furthermore, hiPSCs showed to be also more likely to acquire frequent polymorphic alterations with no pathogenic effect, since we detected this kind of alteration in 14.29% of hiPSCs and only in 1.69% of hESCs (Figure 4B). We also investigated whether different hPSC types would have a proclivity for specific alterations detected by this technique (Figure 4C). Observations showed that the detected karyotypic alterations in hiPSCs tended to be distributed among chromosomes 8 (6.12%), 9 (6.12%), 12 (4.08%), 20 (12.25%) and 21 (8.16%), whereas 75.06% of all karyotypic alterations in hESCs were mostly limited to gains of the chromosome 20 or its long arm (detected in 11.86% of all analyzed hESCs). The frequent structural alteration (46,XX,inv(9)(p11q13)) was the only other aberration detected in hESCs, with a prevalence of 1.69%. Overall, these results highlight a tendency of hiPSC lines to acquire a wider range of numerical and structural chromosomal alterations, both pathogenic and polymorphic, in comparison to hESC lines.

#### 3.2.4. Comprehensive aCGH of hPSCs Reveals Hidden Recurrent Copy Number Variations (CNVs) and Pathological Copy Number Alterations (CNAs)

Since aCGH analysis can detect DNA copy number changes (deletions or duplications) genome wide at a higher resolution than G-banding analysis, we used this cytogenetic technique to refine our analysis. Briefly, we performed aCGH analysis (with a resolution enabling detection of DNA gains or losses as small as 100–350 kb) to verify all previously reported abnormalities found by G-banding, as well as to identify additional submicroscopic unbalanced rearrangements of genetic material which had remained hidden.

As expected, we noticed an increased prevalence of alterations in hPSCs by using this technique (Figure 5A). A total of 47 alterations were detected in 26 analyzed hPSC cultures, from which 34 did not correspond to reported polymorphic variants (which we will refer to as Copy Number Alterations or CNAs from this point), and 13 were considered polymorphic Copy Number Variations (CNVs) which are normal variants in the population. Since some of these alterations were simultaneously present in the same culture (i.e., some cultures presented both CNAs and CNVs), we decided to divide the analyzed hPSC cultures into three groups: (i) genomically stable cultures; (ii) cultures in which only one or more CNVs were detected, but not CNAs; and (iii) cultures which contained at least one CNA, regardless whether or not CNVs were simultaneously detected (that is, cultures containing any alteration with pathogenic potential). Only 7.6% of all analyzed hPSCs (*n* = 26) did not present any genomic alteration. Detection of polymorphic CNVs occurred in 11.5% of these cells, whereas likely pathogenic CNAs were found in 61.5% of hPSCs. A detailed list of detected CNVs and CNAs can be found in Table 3 and Table 4, respectively.

#### 3.2.5. Polymorphic CNVs Are More Prevalent in hiPSCs Than in hESCs, while Potentially Pathogenic CNAs Are More Frequent in hESCs

We then aimed to elucidate what was the contribution of each hPSC type to this frequency distribution. We observed that the relative amount of genomically stable cells was higher in hiPSC (5.88%) than in hESCs (for which surprisingly 100% of the analyzed cells were genomically unstable). While all the alterations found in hESCs were pathogenic, this was found to be the case for 64.86% of the alterations found in hiPSCs, whereas the rest were considered CNVs (35.14%). This shows an increased prevalence of CNVs in hiPSCs, whereas analyzed hESCs seem to have an increased likelihood of acquiring CNAs with potential pathogenic effects, as compared to hiPSCs (Figure 5A).

Next, we evaluated the differences in the propensity to acquire subchromosomal aberrations among different hPSC types. To that aim, we studied whether the probability of hESCs to acquire chromosomal alterations over time was significantly different than that of hiPSCs. We normalized the passage number for both hPSC groups and used Kaplan–Meier curves to show probability of hESCs and hiPSCs to acquire genomic alterations detectable by aCGH at a certain passage number (Figure 5B). The hESCs and hiPSCs were analyzed over a period of 33 and 11 passages in vitro, respectively. Our results showed no significant difference among hPSC groups (*p* = 0.078).

By studying the frequency, types and distribution of detected CNVs and CNAs in more detail we observed that, once again, polymorphic CNVs showed a tendency to be distributed throughout the whole genome (Figure 5C), since they were detected in chromosomes 1 (present in 11.54% of analyzed hiPSCs), 3 (11.54%), 4 (7.69%), 5 (7.69%), 10 (7.69%) and 22 (3.84%) (Table 3). Chromosomes 1 and 3 in hiPSC lines presented a special predisposition to acquire CNVs, having the highest frequencies of this type of alteration. Regarding the differential prevalence of CNAs among hESCs and hiPSCs, we found that hiPSCs also display a more varied set of alterations, which also showed to be more distributed throughout the genome (Figure 5D). In fact, and as shown in Table 4, CNAs in hiPSCs were detected in chromosomes 1 (which was altered in 5.88% of analyzed hiPSCs), 2 (23.52%), 3 (11.76%), 4 (11.76%) 6 (11.76%), 7 (5.88%), 8 (11.76%), 11 (11.76%), 12 (5.88%), 14 (5.88%), 15 (17.64%), 20 (5.88%) and 22 (11.76%), whereas hESCs alterations were limited to chromosomes 10 (where alterations were detected in 44.44% of hESCs), 14 (22.2%) and 20 (66.67%). These data indicate that CNAs are more frequent in hESC lines than in hiPSCs lines, and that most CNAs affect chromosome 20, since 26.92% of all hPSCs (and 66.67% of hESCs) harbored CNAs in this chromosome (which represent around 16.67% of all detected CNAs; Figure 5D). 

We further characterized the genomic alterations found in hPSCs by analyzing the relative abundance of DNA gains and losses within each alteration type (CNVs and CNAs). We found that deletions represented 69.23% of all CNVs. On the other hand, only 30.77% of CNVs (more than 2 times less) corresponded to insertions (Figure 5E). Notably, we observed that gains of genetic material (insertions or duplications) accounted for 56.76% of all detected CNAs, while only the remaining 43.24% were deletions. Thus, these results show that insertions or duplications of genetic material are more likely to cause a potentially pathogenic CNA (frequency increased by 31% compared to deletions for this type of alterations), whereas deletions mostly give rise to CNVs (Figure 5E). 

### 3.3. Analysis on the Effect of QMS Implementation on the Acquisition Rate of Genomic Alterations by hPSCs

#### 3.3.1. The Introduction of SOPs Results in a Reduction in the Prevalence and Types of De Novo Karyotypic Pathogenic Alterations and FPAs in hPSCs Cultured In Vitro

In order to investigate the effect of the introduction of cell culture SOPs under a QMS on the observed prevalence of CNVs and CNAs in hPSCs, only those alterations caused after adapting the hPSC lines to these conditions must be considered. We hypothesized that these alterations corresponded to de novo CNVs or CNAs, which had been acquired because of specific cell culture conditions, and hence must be distinguished from all those alterations which hPSCs already harbored either because they were already present in the embryo or as a consequence of suboptimal culture conditions or hiPSC generation (pre-existing alterations). To that aim, we divided the analyzed hPSCs into two groups: (i) hPSCs which had been adapted and cultured using standardized conditions for at least four passages (post-adaptation, or PoA); and (ii) hPSCs which had not been cultured under these conditions for at least four passages, being expanded using their original protocols (pre-adaptation, or PrA). 

Briefly, we started by identifying all those alterations which had been detected in PrA hPSC lines, and classifying them as inherited variants (that is, alterations which had been originated before adaptation and hence were already harbored by the cells). After that, we studied the prevalence of pre-existing alterations, detected by conventional G-banding, in PrA hPSC lines. Pre-existing polymorphic chromosomal alterations were found to affect 11.76% of PrA hPSCs, while potentially pathogenic karyotypic aberrations were identified in 29.41% of these cells (Table 5).

To identify de novo alterations in PoA hPSCs, we identified all pre-existing alterations which had also been detected in PoA hPSCs. Then, these alterations were filtered out from the analysis, and the remaining genomic alterations were classified as de novo variants. The prevalence of these alterations in PoA hPSCs was then established (Table 6) and compared against that of pre-existing alterations in PrA hPSCs. By using this strategy, we were able to evaluate whether these conditions have a positive impact in preventing the acquisition of karyotypic alterations of hPSCs in vitro. The results showed that the prevalence of de novo polymorphic karyotypic variants in PoA hPSCs was 5.49%, while de novo karyotypic aberrations were only present in 8.79% of the cells (Table 5). This represents a reduction of 53.32% (2.14-fold) and 77.59% (3.35-fold) in the prevalence of polymorphic and pathogenic karyotypic alterations, respectively. These observations highlight that the development and use of hPSC culture SOPs results in a reduction in the frequency at which karyotypic polymorphic variants and aberrations are acquired, and hence improves genomic stability of these cells.

#### 3.3.2. The Use of SOPs also Results in a Reduction in the Prevalence and Types of De Novo CNVs and CNAs in hPSCs Maintained In Vitro

As expected, we also detected a higher prevalence of subchromosomal genomic alterations by aCGH in the analyzed PrA hPSCs, especially regarding potentially pathogenic alterations. More specifically, 46.67% and 80% of hPSCs presented CNV and CNA, respectively before adaptation (Table 5). On the other hand, once adaptation to the new conditions had been completed and under the new culture conditions, only 18.18% and 36.36% of hPSCs were found to be affected by de novo CNVs and CNAs, respectively (Table 5). Hence, in this case we observed a 2.2-fold reduction in the prevalence of de novo CNAs, and a 2.57-fold reduction in the prevalence of de novo CNVs. In addition, we observed that while PrA hPSCs displayed a wide range of different CNVs (6 different types) and CNAs (15 types), genomic alterations in PoA cells were limited to only two different CNVs and three different CNAs (Table 6).

These observations reveal that the application of SOPs to hPSCs maintenance under a QMS leads to more genomically stable cell products at a submicroscopic level. This strategy not only results in a reduction in the prevalence and types of polymorphic genomic variants, but also in a drastic decrease in the acquisition rate of potentially pathogenic karyotypic aberrations and subchromosomal CNAs.

#### 3.3.3. The Standardization of hPSC Maintenance Procedures via QMS Implementation Leads to a Huge Reduction in the Probability of hPSCs Acquiring De Novo Karyotypic Aberrations and Subchromosomal CNAs during In Vitro Culture

Once we had observed that the absolute prevalence and types of both de novo chromosomic and subchromosomal alterations was reduced in hPSCs, which had been adapted to standardized in vitro cell culture conditions, we investigated whether the probability of hPSCs to become genomically altered over time was significantly lower during the PoA period compared to the PrA period. To that aim, we first normalized the passage number for both the PrA and PoA hPSC groups. In the first case, passage number 0 corresponds to the passage number at which each cell line arrived at our laboratory; however, for the post-adaptation group, passage number 0 equals the first passage number at which the adaptation to standardized conditions was completed (that is, the fourth passage number using our cell culture protocol).

Then, we studied the evolution of the probability of acquisition of genomic alterations in hPSCs over time in culture (passage number) for both the PrA and PoA periods. To that aim, we used Kaplan–Meier curves to show what the probability of a given analyzed hPSC acquiring a given type of de novo genomic alteration at a certain passage number for both groups. This analysis was performed for both chromosomal alterations detected by G-banding karyotype (Figure 6A) and subchromosomal alterations detected by aCGH (Figure 6B).

For chromosomal alterations detected by G-banding, PrA and PoA hPSCs were analyzed over a period of 42 passages in vitro (Figure 6A). We found that the probability of de novo chromosomic alterations in PoA hPSCs was significantly lower than that of PrA hPSCs (*p* < 0.0001). In fact, this probability was around 4 times higher (~40% vs. ~10%) for PrA hPSCs at only passage 7 than PoA hPSCs which had been cultured under standardized conditions for around 6 times longer (passage 42). Interestingly, we observed that this difference was mainly due to karyotypic aberrations with pathogenic potential (*p* < 0.0001), for which the probability of being acquired by PrA hPSCs (~30% by passage 7) was around three times higher than that for PoA hPSCs cultured 6 times longer (~10% by passage 42). Although this probability was found to be different for both groups when only considering chromosomal FPAs (*p* = 0.0133), this difference was not as striking (~6% for PrA hPSCs at passage 7 vs. ~5% for PoA hPSCs at passage 42). This highlights that the probability of hPSCs acquiring de novo pathogenic karyotypic aberrations during in vitro culture is hugely reduced (although not completely eliminated) by the adaptation of these cells to standardized conditions.

Finally, for subchromosomal alterations detected by aCGH, we analyzed data from PrA and PoA hPSCs over a more reduced period of time of 21 passages in vitro (Figure 6B). In this case, we also observed a significantly reduced probability (*p* = 0.0006) of acquiring subchromosomal alterations in the hPSCs of the PoA group (~40% by passage 21) compared to hPSCs in the PrA group which had been cultured in our laboratory for about half the time (~80% by passage 10). Once again, this difference was found to be mainly caused by pathogenic CNAs (*p* = 0.0023), since cells in the PrA group showed a 2-fold increase in the probability of acquiring CNAs compared to hPSCs in the PoA group which had been cultured in vitro by our group for two times longer (~80% by passage 10 vs. ~40% vs. passage 21). On the other hand, the probability of cells from both groups to acquire subchromosomal CNVs over time was not found to be significantly different (*p* = 0.3291). Hence, these data also underscore that the probability of hPSCs acquiring de novo CNAs during in vitro culture is hugely reduced (although not completely eliminated) by the adaptation of these cells to standardized conditions after SOP implementation, while the probability of acquisition of polymorphic CNVs is not altered.

## 4. Discussion

Genomic integrity of hPSCs is essential for research and clinical applications. However, genetic abnormalities can accumulate during hPSC routine culture. However, all of the available literature indicates that hPSCs are often genomically unstable with a growing body of research reporting a propensity for genomic instability during prolonged hPSC culture [47,48]. This has posed concerns for basic research and clinical applications. In this context, the standardization of the quality control of hPSCs is becoming increasingly important. This work was prompted by the clear need for studies that not only emphasize the importance of maintaining and monitoring genomic integrity of hPSCs, but also evaluate potential approaches to minimize and better detect this variability. The aim of our study was to investigate the impact of a QMS ensuring standardized cell culture conditions and genomic screening on the genomic stability of hPSCs. 

Our initial observations showed that the number of scientific articles on hPSCs have grown exponentially during the last 24 years, as a possible consequence of the three main breakthroughs of the hPSC field to date: the derivation of the first hESC (1998) [2], the generation of the first hiPSC line (2007) [4] and the start of the first hiPSC-based clinical trial (2013), the RIKEN trial. This last milestone seems to have boosted the interest in hiPSCs by confirming that clinical translation of these cells was a real possibility. Conversely, our data suggest that hESCs, linked to numerous ethical and practical issues, would have started to be seen as less attractive by the community once an alternative hPSC source had reached a clinical setting. In parallel to the unprecedented growth of the field, we also noticed an increasing, yet insufficient number of papers including genomic integrity data of hPSCs experience. In fact, our analysis has revealed that only around 1 out of 20 scientific publications related to hPSCs includes information on the assessment of genomic integrity of the cells being used for basic and translational research. This underlines that hPSC genomic integrity remains largely underreported in scientific publications, and evidence that that most hPSC researchers worldwide may be unknowingly and routinely using hPSCs which have been altered during culture. Interestingly, the number of hPSC publications reporting this kind of information experienced a huge increase around the time period by when the RIKEN hiPSC clinical trial was halted [17]. Considering that the RIKEN trial was precisely stopped after detecting a mutation of unknown significance in the hiPSC-derived cell product to be used for transplantation, this could be a direct consequence of a raised awareness of the importance of proper genomic integrity assessment for the successful clinical translation of hPSCs.

Routine monitoring of hPSC cultures for the appearance of genomic alterations is essential, since some common variants can change from an undetectable state to compose the totality of the culture within five to ten passages [49]. Different screening methods have different advantages and limitations, which are linked to their ability to detect mosaic populations, resolution, the types of variants which are detectable, the cost, the speed and whether specialist facilities and expertise are required [16]. Currently, there is a lack of consensus on a standard strategy for monitoring genomic integrity of hPSCs, which gives rise to different standards being applied in different laboratories. Consequently, clinical translational efforts are threatened, as data reproducibility is hampered. Along these lines, Rossi et al. have recently called for the international stem cell community to put forward clear guidelines on genome monitoring of hPSCs [50]. In this study, we selected a screening strategy which combines a classical cytogenetic technique (karyotypic G-banding) with the molecular technique aCGH. In principle, both karyotyping and arrays are genome-wide technologies which can be used to assess the presence of genomic alterations. The main difference between them is the resolution: while G-banding is only capable of detecting aberrations greater than 5 Mb, aCGH can detect alterations down to 1 kb. The limited resolution of G-banding could have presented an issue for assessing the genetic status of hPSC cultures, as some of the most common changes in hPSCs are present at a subchromosomal level. On the other hand, it is a cost-effective, methodologically simple screening method for larger cytogenetic alterations and can detect numerical and structural aneuploidies as well as transpositions, deletions, duplications, and even inversions of chromosomal arms or larger chromosomal fragments (being the only convenient method to detect balanced genomic rearrangements) [51]. Comparative Genomic Hybridization array, in turn, enables high-resolution detection of submicroscopic unbalanced genomic alterations [52]. Nonetheless, the frequent use of this technique is limited by its higher costs compared to conventional cytogenetic techniques. By integrating molecular and traditional cytogenetics, this comprehensive approach amenable to routine use, enables us to identify most chromosomal and subchromosomal genomic alterations which hPSCs could acquire in culture. However, we cannot exclude that mosaic alterations may have been unnoticed since this strategy is still limited by the low sensitivity of both approaches to low-grade mosaicism. In fact, all the chromosomal aberrations detected in this study were present in mosaic cultures at various degrees. Hence, although steps (including this study) can be taken to minimize the appearance of variants within an hPSC culture, it is of paramount importance to find alternative genomic analysis which can detect low levels of mosaicism within a culture.

In this work, we found that 24.07% of the hPSCs being analyzed showed some aneuploidy which could be detected by G-banding. These data are consistent with the occurrence reported by separate large-scale studies, in which between 12.9% [26] and 34% [22] of hPSC lines were found to acquire chromosome abnormalities over time. As expected, the higher resolution of aCGH led to detection of larger number of alterations in hPSCs (11.5% and 61.5% of hPSCs harboring CNVs and CNAs, respectively). Overall, gain of chromosome material is better tolerated by hPSCs than loss, and hence it is markedly more common in culture adaptation [34], this is also supported by our observations that all chromosomal aberrations that we found corresponded to insertions or duplications. Interestingly, this does not seem to be the case for subchromosomal alterations, since only 56.76% of CNAs were found to be gains in our dataset. Furthermore, we have observed a tendency of hiPSC lines to acquire wider range chromosomal alterations in comparison to hESC lines, which is in agreement with work from other groups, which demonstrate a lower genomic stability of hiPSCs due at least partially to low fidelity of DNA damage repair [53]. In addition, cell reprogramming to generate hiPSC lines could also contribute to this phenomenon due to massive genome remodeling of hiPSCs [54]. In fact, more than one third (35.14%) of the subchromosomal alterations found in hiPSCs correspond to polymorphic CNVs, whereas CNVs were not detected in any of the analyzed hESCs. Remarkably, CNVs tend to increase their prevalence with time in culture, since cultured hiPSCs have been shown to select rapidly against cells harboring them [54]. However, this observation does not imply that hiPSCs show more frequent pathogenic alterations, since our data suggests that CNAs are more prevalent in hESCs (for which CNAs were detected in all analyzed cells) than in hiPSCs (where only around ~60% of cells harbored some CNA).

Our analysis revealed that the most common genomic alteration in our hPSC cultures was a 20q11.21 amplification, a frequent gain of a small, variable region located near the centromere of chromosome arm 20q [55,56]. This is a recurrent alteration which was noted by SNP array analysis in more than 20% of the cell lines in the ISCI study [22,57], and which results in overexpression of Bcl-xL (isoform of BCL2L1). Notably, this alteration confers a growth advantage due to reduced propensity for apoptosis [55,56], reduced dependence on bFGF, delayed differentiation to neural lineages [58] and reduced efficiency of neuroectodermal lineage commitment (caused by perturbation of SMAD and TGFβ signaling) [59]. In our study, we only detected this aberration in 11.11% of hPSCs by G-banding, which can be explained by the fact that this amplification is often below the resolution of current monitoring methods used in research laboratories, most often G-banding karyotyping [16]. Conversely, by aCGH, we report a much larger prevalence of this alteration in hPSCs (23.07%), closer to what was reported by the ISCI study [22]. Other examples of recurrent chromosomal aberration that we found were trisomies of chromosomes 8 and 12. The later alteration results in NANOG overexpression and subsequent increased proliferation rate and replication and reduced propensity for spontaneous differentiation and apoptosis [34]. Interestingly, we only detected this trisomy in hiPSCs (4.08%), a prevalence much lower than what has been reported for hESCs and hiPSCs (42.6% and 31.9%, respectively) [26]. Remarkably, most of these alterations have been associated with various cancers, notably the association of trisomy of chromosome 12 with embryonal carcinoma cells [60] or testicular germ cell tumors [61], or 20q11.21 amplification with colorectal cancer [62], cervical cancer [63] and tumorigenic transformation [64,65]. Regarding subchromosomal alterations besides gains in chromosome 20 and a small insertion in 14q23.3 [66], none of the detected CNAs is, to our knowledge, associated with malignant transformation or impaired functional properties of cultured hPSCs.

Contrary to what is observed in clinical practice and R&D in the pharmaceutical industry, basic and preclinical biomedical academic laboratories often lack a structured QMS such as ISO9001:2015. The implementation of such systems in academia is challenging, mostly due to its immense documentation load and fears that it would limit creativity. However, this lack of QMSs, along with transparency are two reasons for the ongoing reproducibility crisis [67,68,69] and the subsequent translational roadblock. This becomes especially relevant in the case of genomic alterations acquired by hPSCs in vitro (even clinical grade) [35], which can ultimately confound experimental results and potentially jeopardize the outcome of clinical therapies. Our results show that the adaptation of hPSC lines to maintenance conditions described by SOPs under ISO9001:2015 resulted in a notable decrease in the prevalence and types of chromosomal and subchromosomal genomic alterations. In fact, we discovered a specially pronounced reduction of 70.11% in pathogenic chromosomal aberrations in hPSCs after they had been adapted to the new conditions, while similar yet less marked decreases were observed for FPAs (53.31%) and subchromosomal CNAs (54.55%) and CNVs (61.05%) post-adaptation. Our data also demonstrated a decreased probability to acquire pathogenic genomic alterations of hPSCs cultured under the new standardized conditions. In fact, the probability of pathogenic chromosomal alterations in hPSCs was found to be decreased by three times under these culture conditions, even when hPSCs had been cultured for an in vitro period six times longer compared to cells cultured under non-standard conditions. Similarly, the use of SOPs under ISO9001:2015 resulted in two times less probability of cells to acquire de novo pathogenic subchromosomal CNAs even when time these cells had been maintained in vitro for twice as long as the controls. Our observations provide evidence of the positive impact of a QMS in the genomic stability of hPSCs maintained in vitro, and thus support the implementation of such systems in academic laboratories performing hPSC research, to enhance the reproducibility of experimental results.

We found that the most abundant de novo pathogenic alterations acquired by hPSCs under ISO9001:2015 were an insertion in 14q23.3 (which has been associated to a cancerous phenotype) [66], followed by an insertion in 14q32 (linked to predisposition to early clonal hematopoiesis leading to myeloid neoplasms) [70], the 20q11.21 amplification and trisomies in chromosomes 8 and 12. Given that all of these alterations confer a selective advantage to variants which could lead to malignant phenotypes, it is clear that despite the improved genomic stability of hPSCs, genetic alterations which may potentially jeopardize the safety or impair the function of hPSCs are not completely removed from cultured cells. We hypothesize that this may be a consequence of adapting hPSC lines to standard conditions well after they have been derived. It is still to be determined if implementation of a QMS since derivation will result in a greater improvement of hPSC genomic stability. Regardless, the frequent genomic monitoring of hPSCs resulting from the application of a QMS enhances the ability of researchers to detect a larger number of genomic alterations earlier, before they hPSC variants take over the cultures.

## 5. Conclusions

This work demonstrates that application of a QMS such as ISO9001:2015 to standardize cell culture conditions and genomic monitoring routines via the development and use of SOPs which lead to a striking improvement of genomic stability in hPSC lines cultured in vitro. These results serve as a proof of concept to support the implementation of QMS in laboratories performing hPSC research. Furthermore, upon noticing that information on hPSCs is largely unreported in the scientific literature, we provide evidence that genomic integrity of cultured hPSCs should be frequently monitored using a combination of cytogenetic and molecular screening methods. Overall, we hypothesize that implementation of this strategy will be key to overcome the current reproducibility crisis, hence ensuring a smooth translation of hPSC-based technologies.

## Figures and Tables

**Figure 1 cells-11-01984-f001:**
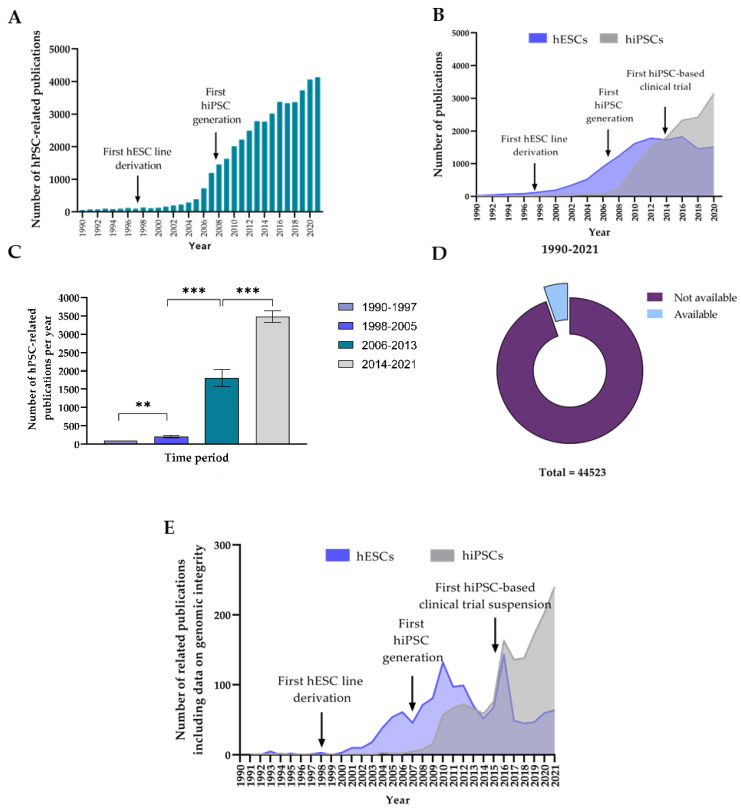
Analysis of the evolution in number of human pluripotent stem cell (hPSC)-related publications and level of reporting of hPSC genomic integrity monitoring: (**A**) Bar graph representing the number of publications on hPSCs per year; (**B**) compound line graph of trend of publications per year during on human embryonic stem cells (hESCs) and human induced pluripotent stem cells (hiPSCs). Notice the three inflection points relative to three major events in the PSC field: the first hESC line derivation by Thomson in 1998 [2], the first hiPSC generation by Takahashi and Yamanaka in 2007 [4] and the RIKEN first hiPSC-based clinical trial in 2013; (**C**) histogram of average number of publications per year in distinct temporal windows; (**D**) publications of hPSC research containing information on genomic integrity monitoring of hPSCs. Available segment indicates the percentage of articles with published data about genomic integrity of hPSCs, where the mean ± the standard error of the mean (SEM) is showed; (**E**) trend of publications per year on hESCs and hiPSCs reporting information on genomic integrity of cells used for the study: increasing interest about monitoring genomic integrity follows critical events about their usage.

**Figure 2 cells-11-01984-f002:**
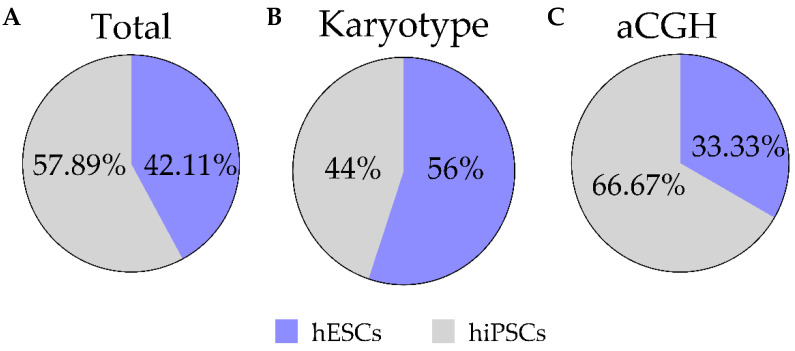
Descriptive statistics of the analyzed dataset. Pie charts showing the relative amount (%) of hESC and hiPSC lines used in this study (**A**), as well as the proportion of G-banding (**B**) and CGH array (aCGH) (**C**) which were performed in each of the hPSC types.

**Figure 3 cells-11-01984-f003:**
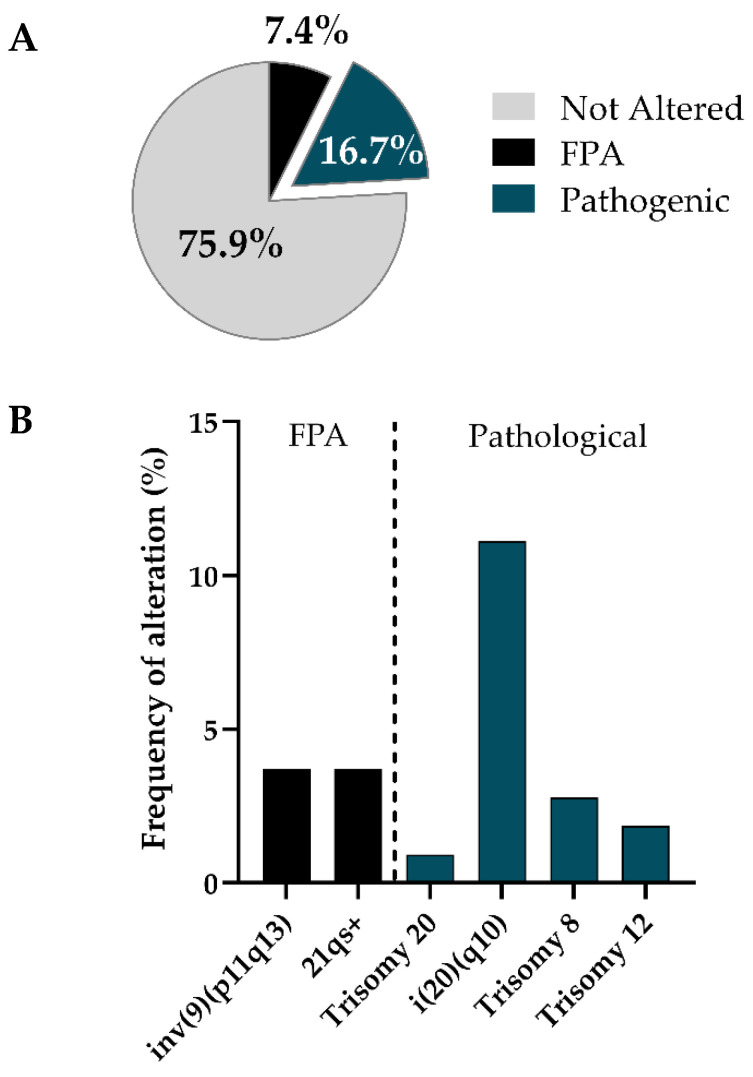
Recurrent karyotypic aberrations are detected in cultured hPSCs by G-banding: (**A**) Pie chart of fraction of genomic alterations detected with karyotype on the total of the cell lines analyzed; (**B**) histogram of chromosomal abnormalities found in hPSCs by using karyotype. Each bar represents the percentage of time that the alterations has been found in hPSCs. FPA: Frequent Polymorphic Alterations.

**Figure 4 cells-11-01984-f004:**
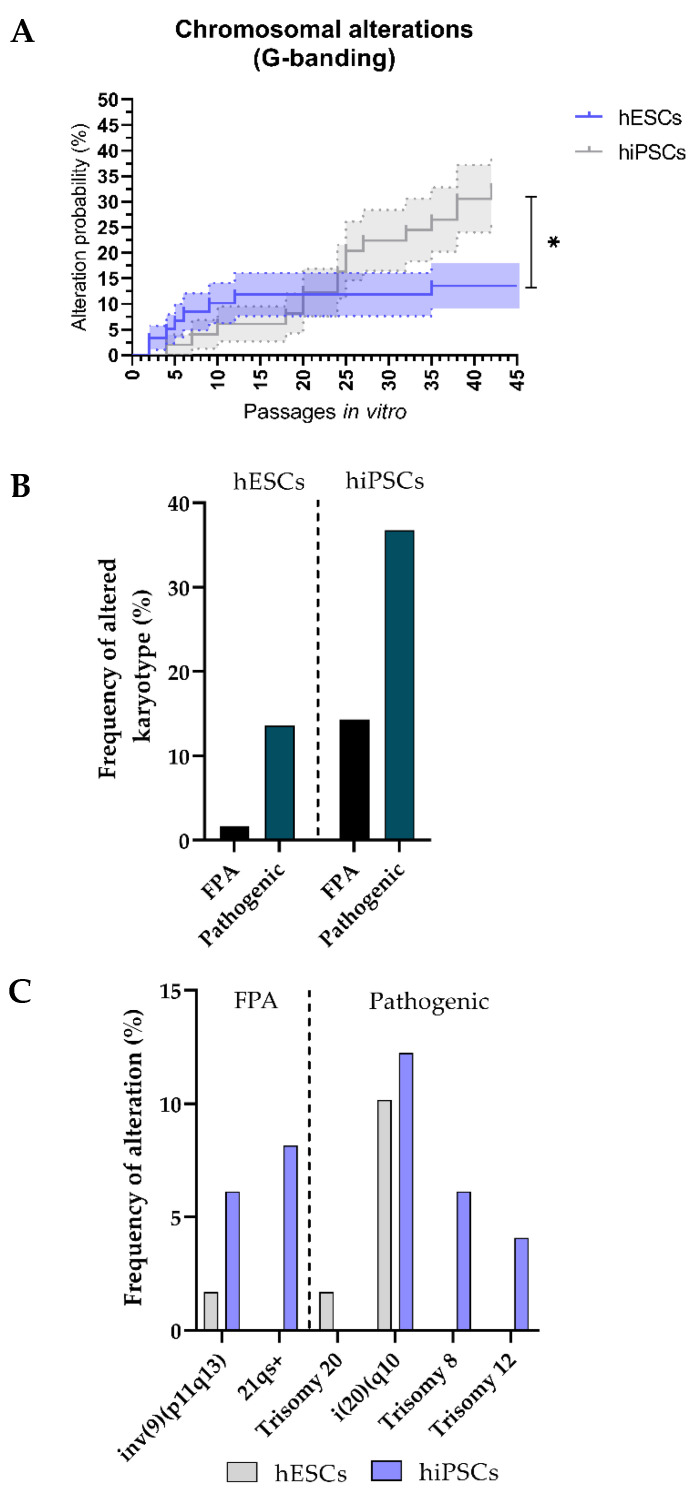
The hiPSCs lines show a higher prevalence of karyotype genomic aberrations detected by G-banding as compared to hESCs: (**A**) Survival analysis (Kaplan–Meier curves) of the evolution of the probability (%) of hESCs and hiPSCs being affected by chromosomal genomic alterations detected by G-banding over time (passages in vitro). Error bars represent standard error of the mean (s.e.m.); (**B**) bar graph of percentage of frequency of all alterations found in hESCs compared with hiPSCs on a total number of 108 karyotypes performed; (**C**) histogram showing the frequency at which different types of FPAs and pathologic alterations are detected in hESCs and hiPSCs via karyotype. FPA: Frequent Polymorphic Alterations.

**Figure 5 cells-11-01984-f005:**
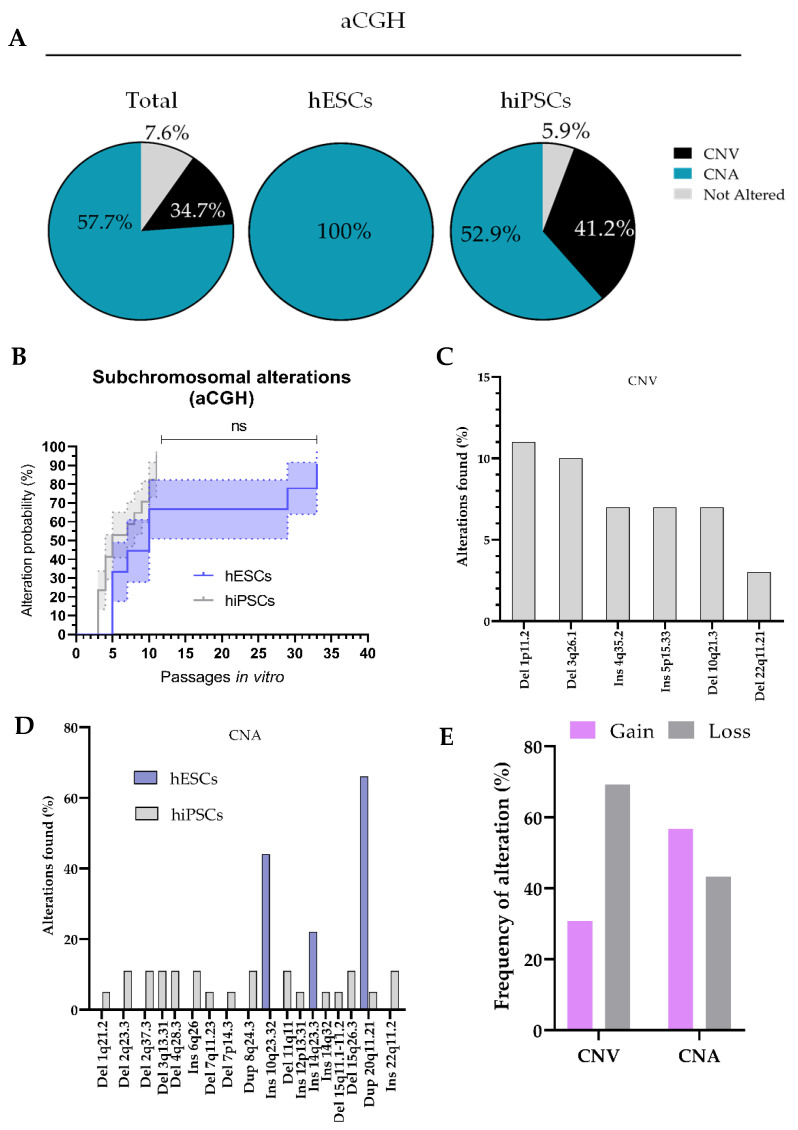
Recurrent subchromosomal Copy Number Variations (CNVs) and pathogenic Copy Number Alterations (CNAs) are detected in cultured hESC s and hiPSCs by Comparative Genomic Hybridization CGH array (aCGH) analysis: (**A**) Pie charts showing the percentage of cells harboring Copy Number Variations (CNVs) or Copy Number Alterations (CNAs), as well as cell lines for which no alteration were detected; (**B**) survival analysis (Kaplan–Meier curves) of the evolution of the probability (%) of hESCs and hiPSCs being affected by subchromosomal genomic alterations detected by aCGH over time (passages in vitro). Error bars represent standard error (S.E.M.); (**C**,**D**) histograms showing the frequency at which different types of CNVs (**C**) and CNAs (**D**) are found in analyzed hESCs and hiPSCs; (**E**) bar graph showing the frequency at which DNA gains and losses are detected with aCGH for both CNV and CNA group.

**Figure 6 cells-11-01984-f006:**
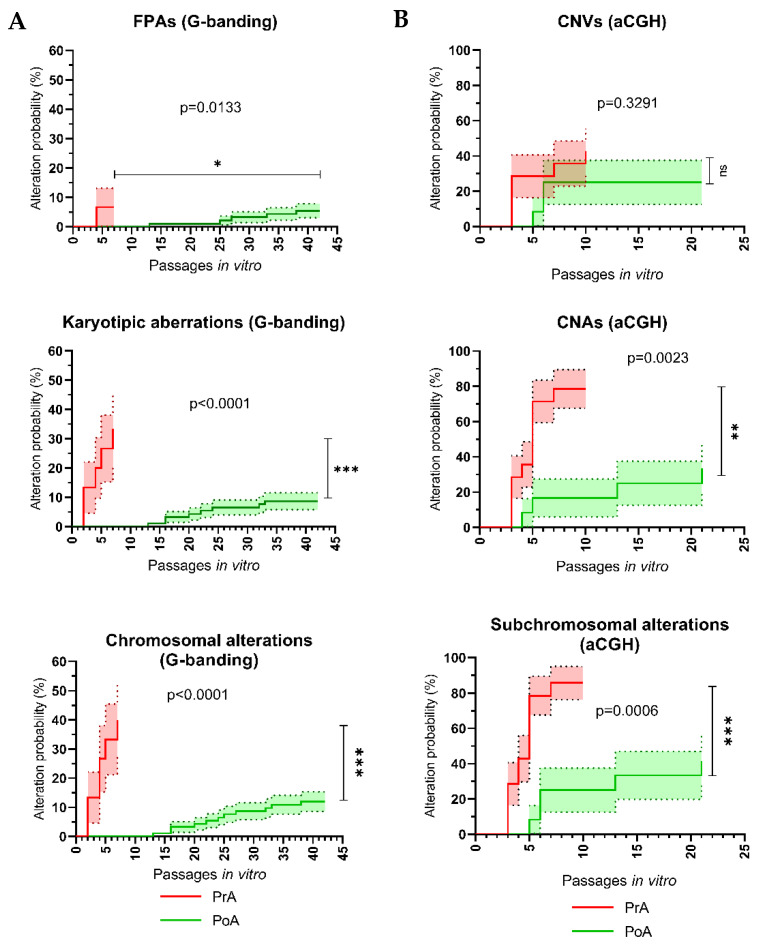
Adaptation to standardized cell culture conditions described by standard operating procedures (SOPs) under a Quality Management System (QMS) results in improved genomic stability of cultured hPSCs. Survival analysis (Kaplan–Meier curves) of the evolution of the probability (%) of hPSCs being affected by de novo genomic alterations over time (passages in vitro) before and after adaptation to standardized cell culture conditions, using G-banding karyotyping (**A**) and Comparative Genome Hybridization array (aCGH; **B**) as detection methods. For each panel, the top, middle and bottom plots show the probability of hPSCs acquiring polymorphic alterations, pathogenic alterations and total alterations, respectively. Pre-adaptation (PrA) and post-adaptation (PoA) refer to hPSCs analyzed before and after completing adaptation to the standard cell culture conditions used in our laboratory, respectively. Error bars represent standard error (S.E.M.). FPA: Frequent Polymorphic Alteration; CNV: Copy Number Variation; can: Copy Number Alteration.

**Table 1 cells-11-01984-t001:** List of human pluripotent stem cell (hPSC) lines used for this study.

Cell Line	hPSCreg^®^/NIH ID	Disease Status	hESC/hiPSC	Institution *
GENEA-019	GENEAe020-A	Unaffected	hESC	GENEA
H9	WAe009-A	Unaffected	hESC	WCRI
RUES2	RUESe002-A	Unaffected	hESC	RU
RUES2 2123	RUESe002-A-1	Unaffected	hESC	RU
CTR2161	CHDIi013-A	Unaffected	hiPSC	CHDI
CTR2162	CHDIi014-A	Unaffected	hiPSC	CHDI
CTR2164	CHDIi016-A	Unaffected	hiPSC	CHDI
CTR2175	CHDIi027-A	Unaffected	hiPSC	CHDI
CTR2190	CHDIi042-A	Unaffected	hiPSC	CHDI
CTR33n1	CS83iCTR33n1	Unaffected	hiPSC	CSRMI
GENEA-018	NIHhESC-12-0169	Affected HD	hESC	GENEA
GENEA-020	GENEAe015-A	Affected HD	hESC	GENEA
HD65Q	NA	Affected HD	hESC	TLGM
RUES2 1811	RUESe002-A-2	Affected HD	hESC	RU
HD60n5	CS21iHD60n5	Affected HD	hiPSC	CSRMI
HD2158	CHDIi010-A	Affected HD	hiPSC	CHDI
HD2170	CHDIi022-A	Affected HD	hiPSC	CHDI
HD2174	CHDIi026-A	Affected HD	hiPSC	CHDI
HD2197	CHDIi049-A	Affected HD	hiPSC	CHDI

* GENEA: Genea Biocells, Australia; WCRI: WiCell Research Institute, USA; RU: The Rockefeller University, USA; CHDI: Cure Huntington’s Disease Initiative Foundation Inc., USA; CSRMI: Cedars-Sinai Regenerative Medicine Institute, USA; TLGM: Translational Laboratory in Genetic medicine, Singapore.

**Table 2 cells-11-01984-t002:** List of chromosomal alterations detected by G-banding in human embryonic stem cell (hESC) and human induced pluripotent stem cell (hiPSC) lines.

Alteration		% of Affected hPSCs	% of Affected hESCs	% of Affected hiPSCs
Chromosomal pathogenic alterations	47,XX,+8	2.78	NA	6.12
	47,XX,+12	1.85	NA	4.08
	46,XX,i(20) (q10)	11.11	10.17	12.24
	47,XX,+20	0.93	1.69	NA
Frequent Polymorphic Alterations (FPAs)	46,XX, inv(9) (p11q13)	3.70	1.69	6.12
	46,XX,21qs+	3.70	NA	8.16

**Table 3 cells-11-01984-t003:** List of Copy Number Variations (CNVs) detected by CGH array (aCGH) in hPSCs.

Alteration	Region	hESC/hiPSC	% Cells Affected	% of Total CNVs
Deletion	1p11.2	hiPSC	11.54	23.08
Deletion	3q26.1	hiPSC	11.54	23.08
Insertion	4q35.2	hiPSC	7.69	15.38
Insertion	5p15.33	hiPSC	7.69	15.38
Deletion	10q21.3	hiPSC	7.69	15.38
Deletion	22q11.21	hiPSC	3.85	7.69

**Table 4 cells-11-01984-t004:** List of Copy Number Alterations (CNAs) detected by aCGH in hESCs and hiPSCs.

Alteration	Region	hESC/ hiPSC	% Cells Affected (All hPSCs)	% of Cells Affected (hPSC Type)	% of Total CNAs (All hPSCs)	% of Total CNAs (hPSC Type)
Deletion	1q21.2	hiPSC	3.85	5.88	2.70	4
Deletion	2q23.3	hiPSC	7.69	11.76	5.41	8
Deletion	2q37.3	hiPSC	7.69	11.76	5.41	8
Deletion	3q13.31	hiPSC	7.69	11.76	5.41	8
Deletion	4q28.3	hiPSC	7.69	11.76	5.41	8
Insertion	6q26	hiPSC	7.69	11.76	5.41	8
Deletion	7q11.23	hiPSC	3.85	5.88	2.70	4
Deletion	7p14.3	hiPSC	3.85	5.88	2.70	4
Duplication	8q24.3	hiPSC	7.69	11.76	5.41	8
Insertion	10q23.32	hESC	15.38	44.44	10.81	16
Deletion	11q11	hiPSC	7.69	11.76	5.41	8
Insertion	12p13.31	hiPSC	3.85	5.88	2.70	4
Insertion	14q23.3	hESC	7.69	22.22	5.41	8
Insertion	14q32	hiPSC	3.85	5.88	2.70	4
Deletion	15q11.1–11.2	hiPSC	3.85	5.88	2.70	4
Deletion	15q26.3	hiPSC	7.69	11.76	5.41	8
Duplication	20q11.21	hESC and hiPSC	26.92	66.67 (hESCs), 5.88 (hiPSCs)	16.22	24 (hESCs), 4 (hiPSCs)
Insertion	22q11.2	hiPSC	7.69	11.76	2.70	8

**Table 5 cells-11-01984-t005:** Prevalence of chromosomal and subchromosomal alterations detected by G-banding and aCGH, respectively, in pre- and post-adaptation hPSC lines. The reduction in the prevalence of each de novo alteration time in the post-adaptation (PoA) group compared to the pre-adaptation (PrA) group is showed, as well as the fold-change.

Culture Conditions	Chromosomal Alterations (G-Banding)		Subchromosomal Alterations (aCGH)	
	FPA	Pathogenic	CNV	CNA
Pre-adaptation (PrA)	11.76%	46.67%	46.67%	11.76%
Post-adaptation (PoA)	5.49%	18.18%	18.18%	5.49%
$\mathrm{Reduction}\;\left(-\frac{PoA-PrA}{PrA}*100\right)$	−53.31%	−61.05%	−61.05%	−53.31%
Fold-change $\left(-\frac{PrA}{PoA}\right)$	−2.14	−2.57	−2.57	−2.14

**Table 6 cells-11-01984-t006:** List of de novo chromosomal (FPAs and pathogenic alterations) and subchromosomal (CNVs and CNAs) detected by G-banding and aCGH, respectively, in post-adaptation hPSCs.

Alteration		Region	Size	% of Affected hPSCs
Pathogenic chromosomal alterations	Trisomy	+8	NA	3.30
	Trisomy	+12	NA	2.20
	Isochromosome	i(20) (q10)	NA	3.30
Chromosomal FPAs	Pericentric inversion	inv(9) (p11q13)	NA	1.10
	Microsatellite	21qs+	NA	4.40
Subchromosomal CNVs	Deletion	3q26.21	62 kb	9.09
	Insertion	4q35.2	219 kb	9.09
Subchromosomal CNAs	Insertion	14q23.3	370 kb	18.18
	Insertion	14q32	136 kb	9.09
	Duplication	20q11.21	4.7 Mb	9.09

## Data Availability

Not applicable.

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
