# Peer review of "Standardization of Cell Culture Conditions and Routine Genomic Screening under a Quality Management System Leads to Reduced Genomic Instability in hPSCs"

_cells, 2022, doi:10.3390/cells11131984_

Round 1

Reviewer 1 Report

This manuscript stresses out the importance of QMS applications to standardize pluripotent stem cell culture conditions and improve their genomic stability in vitro. It is a very interesting work. However, I have a few small concerns, as follows.

1. Lines 174-176: Suddenly the font is bigger.

2. Line 182: Please, rephrase the following sentence: “Medium was daily changes”.

3. The authors should explain why they only employed female cell lines.

4. The authors should use mL instead of ml everywhere in the manuscript.

5. Lines 232-233: The product abbreviation and information “Dulbecco’s (D)-phosphate-buffered saline (PBS) without calcium and magnesium (D-PBS−/−, 14,190–250; Thermo Fisher Scientific Inc., MA, USA)” was already used above. From this stage on, only the abbreviation should be used.

6. Lines 377-379: The authors stated for examples that “…we found that 24.1% of the hPSCs being analyzed showed genomic alteration which could be detected with the resolution of the used cytogenetic technique (that is, abnormalities above 5-10 Mb), whereas 75.9% of the hPSCs seemed to be unaltered”. I wonder whether the authors normalized those data to passage number. In other words, was the passage number more than the line per se the cause of genomic alterations?

7. Did the authors try to rescue the lines by picking single colonies and analyze whether the abnormalities accumulated already in all the cells of the population? I saw a similar kind of experiment was performed later. Still I wonder whether it is possible to rescue the cell lines if genomic instability is checked regularly. 

8. Lines 428-430: The statement “Overall, these results highlight a tendency of hiPSC lines to acquire a wider range of numerical and structural chromosomal alterations, both pathogenic and polymorphic, in comparison to hESC lines” is very interesting. Again, it would be important to show the correlation with passage number. This should apply to all the remaining analyses.

9. Line 549-550: The authors stated “The use of SOPs also results in a reduction of the prevalence and types of de novo CNVs and CNAs in hPSCs maintained in vitro”. I believe their SOPs are described in the Materials and Methods section. What about the non-SOPs they used as a comparison? Also, how many different non-SOPs did they use, and were all similarly bad?

10. Finally, there are a few typos all over the manuscript, such as “loses” instead of “losses” (line 83), misplaced comma (line 104), “0.2μg/ml” instead of “0.2 μg/mL (line 208), “no not” instead of “to not” (line 215), “0.075M” instead of “0.075 M” (line 218), parenthesis missing (line 275), and others.

Author Response

Dear reviewer,

Thank you very much for your suggestions that will help to improve our manuscript. Here bellow you can find a detailed response to your comments.

All the best

Josep M Canals, on behalf of all the authors.

  1. Lines 174-176: Suddenly the font is bigger.

We have modified the font size

  1. Line 182: Please, rephrase the following sentence: “Medium was daily changes”.

We have substituted the sentence by: The medium was changed every day.

  1. The authors should explain why they only employed female cell lines.

Unfortunately, we don’t have a scientific explanation on why the cell lines we have in the laboratory are derived from females. We had few male cell lines in the lab and they did not expand properly. However, the number of male-derived cell lines we got was very low to get any conclusion on differences between cell’s sex.

  1. The authors should use mL instead of ml everywhere in the manuscript.

We have changes all ml by mL.

  1. Lines 232-233: The product abbreviation and information “Dulbecco’s (D)-phosphate-buffered saline (PBS) without calcium and magnesium (D-PBS−/−, 14,190–250; Thermo Fisher Scientific Inc., MA, USA)” was already used above. From this stage on, only the abbreviation should be used.

We have used the abbreviation a lines 232-233.

  1. Lines 377-379: The authors stated for examples that “…we found that 24.1% of the hPSCs being analyzed showed genomic alteration which could be detected with the resolution of the used cytogenetic technique (that is, abnormalities above 5-10 Mb), whereas 75.9% of the hPSCs seemed to be unaltered”. I wonder whether the authors normalized those data to passage number. In other words, was the passage number more than the line per se the cause of genomic alterations?

No, we did not normalize data in figure 3. However, following referee comments, we have now modified figures 4 and 5 to introduce normalization of G-banding and aCGH by passage number, respectively. The results of these figures are now introduced in the text: lines 499-507 for figure 4 – G-banding; and lines 541-548 for figure 5 - aCGH.

  1. Did the authors try to rescue the lines by picking single colonies and analyze whether the abnormalities accumulated already in all the cells of the population? I saw a similar kind of experiment was performed later. Still I wonder whether it is possible to rescue the cell lines if genomic instability is checked regularly. 

This is an excellent idea to be tested in future projects but unfortunately, we did not try to rescue the genotype by picking single colonies. As we indicated in the manuscript this is a retrospective analysis of the results from the cell lines we cultured in the lab during the last 5 years.

  1. Lines 428-430: The statement “Overall, these results highlight a tendency of hiPSC lines to acquire a wider range of numerical and structural chromosomal alterations, both pathogenic and polymorphic, in comparison to hESC lines” is very interesting. Again, it would be important to show the correlation with passage number. This should apply to all the remaining analyses.

As we explained in response for question 6, we have now introduced the correlation by passage number for G-banding (Figure 4) and for aCGH (Figure 5).

  1. Line 549-550: The authors stated “The use of SOPs also results in a reduction of the prevalence and types of de novo CNVs and CNAs in hPSCs maintained in vitro”. I believe their SOPs are described in the Materials and Methods section. What about the non-SOPs they used as a comparison? Also, how many different non-SOPs did they use, and were all similarly bad?

We have now clarified the number of pre-adaptation protocols (non-SOPs; 5) we used and introduce the references of the original articles of these protocols (179-184).

  1. Finally, there are a few typos all over the manuscript, such as “loses” instead of “losses” (line 83), misplaced comma (line 104), “0.2μg/ml” instead of “0.2 μg/mL (line 208), “no not” instead of “to not” (line 215), “0.075M” instead of “0.075 M” (line 218), parenthesis missing (line 275), and others.

We corrected these typos and reviewed all the text.

Reviewer 2 Report

The manuscript by Molina-Riuz and colleagues describes the results of a large retrospective analysis of the genome integrity of several human ES and iPS cell lines cultivated in mTesR1. Genome integrity is assessed by G-banding karyotype and CGH array. The study concludes that standardization of cell culture conditions and regular genomic monitoring significantly improves genetic stability and genome integrity.

This study is very important for the scientific community working with human ES and iPS cells. Readers will find a wealth of information. The introduction could be shortened but I leave that to the editor's discretion.

I recommend publication.

Author Response

Dear reviewer,

Thank you very much for your positive comments. As the editor did not suggest to reduce the introduction, we prefer to keep our original version.

All the best

Josep M Canals, on behalf of all the authors.